# The Right of Religious Freedom in Light of the Coronavirus Pandemic: The Greek Case

**George Androutsopoulos**

Athens Law School, National and Kapodistrian University of Athens, 106 72 Athens, Greece; georgeand@law.uoa.gr

**Abstract:** The purpose of this article is to take into consideration the impact of unprecedented restrictions due to COVID-19 on the exercise of religious freedom according to the Greek legislation and case-law. The crucial fact to be examined is the proportionality of the exceptional measures of the Greek State. At the beginning of the pandemic, religious ceremonies were allowed only in the presence of clerics, but nowadays they are permitted on the condition that the measures of "social distancing" are being followed strictly. As it is generally accepted, the Greek State managed to deal with the pandemic without deviations from constitutional order and protection of fundamental rights, in accordance with a "pressing social need". In this context, the case-law of the Greek courts is of great importance, which ruled that the above mentioned restrictions did not offend the principle of proportionality, especially because of their temporary and short-term character. Nevertheless, these restrictive measures must be revised from time to time, considering the updated, epidemiological data in order to be selected the most appropriate and less stringent on a case-by-case basis. Consequently, these judgments do not give government a blank cheque regarding the management of the pandemic, but rather provide them with a clear framework which is able to guarantee the measures' accordance with the Greek Constitution. However, the potential risk that people may become used to the restrictions imposed after the crisis has passed must not be overlooked.

**Keywords:** freedom of religion; Greek case-law; COVID-19



## 1. Introduction

It would have been impossible for the Greek legal order to remain unaffected by the COVID-19 pandemic. More precisely, the management of the pandemic crisis has affected conventional relations either directly or indirectly (Karampatzos 2020; Tsolakidis 2020) and inevitably caused the broader decline of both individual rights and fundamental freedoms (Contiades 2020, p. 89ff). Religious freedom, one of the first individual rights claimed in Europe as early as in the 16th century (Konidaris 2020, p. 65) and analyzed into freedom of religious consciousness and freedom of religious worship, is included among them. In particular, the latter refers to the exercise of religious duties[1] in ritualistic forms (Chryssogonos and Vlachopoulos 2017, p. 323) and may occur individually or in community with others, in private or in public, in specially designated houses of worship or outdoors (see for details Papageorgiou 2012, p. 52ff). Likewise, freedom of religion is one of a suite of rights contained in international declarations and conventions, which overlap and interconnect (Ferrari et al. 2020). Whether the strict state/legislative restrictions imposed on the exercise of religious freedom (see also Weiner et al. 2020; Consorti 2020, p. 15) fulfilled the terms and requirements of the condition of proportionality, which ultimately constitutes the threshold of their constitutionality, is a crucial matter. In this case, the role of the courts stands exceptionally strong as "*it seems easy to predict that waves of*

---

[1] For Orthodox Christians, the religious duties are, for example, the following: (a) going to church and take the sacrament (Eucharist), (b) individual or private prayer in the church, (c) listening to the Bible, etc.

*litigation revolving around anti-Covid-19 regulations will flood many courtrooms . . . "* (Pin 2020). Moreover, "*the courts have played an active role in monitoring the executive in many countries to ensure that checks and balances remain robustly in place*" (Tew 2020); hence, it is imperative to combine "*normative and political wisdom*" (Benítez 2020) when issuing their decisions and that because "*in a state of emergency, the judiciary fulfils three main functions: it resolves individual disputes over emergency policies, checks the executive, and clarifies the likely imperfect emergency policies*" (Petrov 2020, p. 80).

## 2. The Legislative Framework

Within the framework of reducing the spread of COVID-19, the Greek Government was forced to legislate emergency measures. Therefore, at first, an Act of Legislative Content (submitted to the Parliament for approval: Law 4682/2020) was issued on the 25th of February 2020 under the title "*Urgent measures to prevent and limit the spread of coronavirus*" through which "*the temporary prohibition of the operation [ . . . ] of places of worship*" was foreseen, amongst others, as a potential measure as decided by the competent Minister of Education following the consultation of the National Commission of Protection of Public Health. In fact, it is taken for granted that the relevant decision, which is made in view of the constitutional principle of proportionality, ought to clearly state, among other things, the specific public health emergency that imposes the issue of the measure as well as to specify the duration of its implementation.

In accordance with the implementation of the aforementioned legislative provision, a ministerial decision (no. 2867) was issued on the 16th of March 2020 which established the temporary prohibition of exercising all kinds of religious services and ceremonies in all places of worship without exceptions as precautionary measures of public health for the period of time from 16 to 30 March. On the contrary, attendance was allowed only in case of individual prayer[2], services were strictly restricted to television or radio broadcasts and funerals were allowed with the presence of a priest and the utmost close relatives. Subsequently, the relevant framework was partly modified, mostly because of Easter, as it allowed gatherings of worship to be strictly held only by the priests and the necessary auxiliary staff (such as preachers and vergers[3]).

After having taken into consideration the country's reduced number of cases of COVID-19, a ministerial order (no. 29519) was issued on the 12th of May 2020 which initially enabled the limited operation of places of worship until the 5th of June 2020 for the first time after the launch of the lockdown, with the provision of a maximum number of attendees inside the buildings as well as the obligation that all the necessary precautionary measures are satisfied as established by the National Commission of Protection of Public Health to avoid the spread of COVID-19.

In accordance with the aforementioned, with a subsequent ministerial order (no. 48967) as issued on the 31st of July 2020, the mandatory usage of non-medical masks was imposed on all places of religious worship regardless of their legal status, not only during services, worship gatherings, holy ceremonies, sacred mysteries, funerals and their relevant events, but also during attendance for individual or private prayer. Finally, the same aforementioned imperatives related to the protection of public health aiming to address the risk of COVID-19 spread imposed the issue of a new ministerial order (no. 50451) on the 9th of August 2020, with regard to the 15th of August, which suspended the conduction of all religious ceremonies regardless of religion or doctrine that are held via a religious procession outside the religious building. In any case, religious buildings have operated without public attendance since the 6th of November 2020.

---

[2]　The potentiality of individual prayer was abolished with ministerial order no. 20036 issued on 22 March 2020 since *no provision was made for the consideration of an exceptional reason for individual movement.*

[3]　A person who takes care of the interior of a church and acts as an attendant during services.

### 3. The Orthodox Church's Reaction

The effectiveness of these restraining measures depended greatly on the cooperation of the Greek Orthodox Church. Before all holy ceremonies in places of worship were temporarily suspended by government decision, the Orthodox Church had decided, on the one hand, that churches would remain open for personal prayer and, on the other hand, that the holy Mass would take place in a modest way with public attendance from Sunday the 22nd of March 2020 until Lazarus Saturday on 11th of April 2020, from 7:00 to 8:00 a.m. It has been supported that *"the Holy Synod on 16 of March resolved to adopt measures of doubtful utility and poor effectiveness for the current level of spread of the epidemic"* (Martinelli 2020, p. 86). However, it is a fact that at that particular moment the Church with its attitude towards the matter had reached the fullest extent of any room for compromise.

Subsequently, the government measure regarding the closure of churches that undoubtedly restricted the religious worship triggered the Greek Orthodox Church which immediately took the initiative in order for the relevant legislative framework to be amended towards finding a less restrictive measure that would allow the freedom of religious worship and protect the public health alike. Therefore, it was requested among others to the State *"to allow the Divine Liturgy and the Mass to be conducted by only one priest without the attendance of the faithful crowd"*.

At this point, the fact must be underlined that the collective exercise of the freedom of worship, which occurs in the churches, is for the Orthodox Church of great importance. Specifically, the Eucharist, where the Christians receive God's gift of himself to them, is the center of Church's life and, in consequence, the participation of Christians in the eucharistic service is the principal event. In such a meaning, a priest cannot perform per Orthodox "canon" law a eucharistic service only by himself, which can be performed in the presence of at least the preacher and verger. Thus, the operation of church buildings closed for public worship only by the implementation of "ecclesiastical economy"[4] could be justified.

Following its meeting on the 1st of April 2020 in order to reevaluate the effects the spread of COVID-19 had on the Church, the Holy Synod took the decision to accept the government's will regarding the "reopening" of the churches especially during the Holy Week "behind closed doors" and with only a priest, a preacher and a verger being present, as delivered to the media by the competent Minister of Education. In the context of the enforcement of the ministerial decision, the Greek Orthodox Church issued a synodic circular (no. 3018/7.4.2020), which notified the Metropolitans to adhere to "the arrangements" in order to prevent any *"malicious comments and blaming"*.

Therefore, except from a few inevitable but altogether reprehensible dissonances[5], the Greek Church finally came in terms with the experts' suggestions, partaking in its own way in the State's attempt to mitigate the spread of COVID-19.[6]

### 4. The Principle of Proportionality

It is widely accepted that the allowed space of restricting the exercise of such a fundamental right as religious freedom, and especially regarding the aspect of freedom of worship, is defined by the adequacy and necessity of the taken measure, as well as its appropriateness towards the desired objective. It is of note that the principle of proportionality can be traced back to the ancient Greek ideal of the principle of "mediocrity", according to which "μέτρον ἄριστον", and was further integrated into the Greek Constitution during its

---

4    The sui generis institution of "ecclesiastical economy" is found only in the field of Ecclesiastical Law and it consists of the non-precise implementation of a particular holy canon on the part of a body of ecclesiastical power in a particular case and for reasons of equity (Papageorgiou 2012, p. 28).

5    Despite the fact that the Greek Orthodox Church notified the Metropolitans to adhere to the aforementioned legislative provisions, some of the Metropolitans refused to apply the measures taken on the grounds that they were hostile to the Orthodox Church. It is obvious that this reaction was unacceptable.

6    The only exception was the Greek Orthodox Church's official refusal to apply the restrictions imposed suddenly and completely barred churches from meeting indoors during *"Theophany"*—a Christian feast day that celebrates the revelation of God incarnate as Jesus Christ (January, 6). It is of note that, as far as I am aware, for first time the Greek Orthodox Church reached the Council of State in globo against the prohibition of conducting holy activities during *"Theophany"*.

revision in 2001 (article 25 paragraph 1 subparagraph δ'), despite the fact that the principle had already been configured by the case-law. In practice, it is probably the most important prerequisite every restriction of a fundamental right should adhere to.

Therefore, its implementation follows three stages of scrutiny; the restriction, thus, must be (a) adequate so as to produce the desired result, (b) necessary, that is a sine qua non condition which means that *"no other option is possible or that the consequences will be dire if the restriction is not imposed"* (Gunn 2012, p. 261) and *"the necessity must be of such a gravity as to trump the exercise of religious liberty"* (Hill 2020). In other words, *"the test of necessity asks whether the decision, rule or policy limits the relevant right in the last intrusive way compatible with achieving the given level of realization of legitimate aim. This implies a comparison with alternative hypothetical acts (decisions, rules, policies etc.), which may achieve the same aim to the same degree but with less cost to rights"* (Rivers 2006, p. 198) and (c) proportionate (stricto sensu or striking a balance between competing rights and interests in a particular case: Anđelković 2017, p. 241), meaning that the burden caused by the restriction of freedom must not exceed the positive outcome for the public interest caused by restriction (Robbers 2005, pp. 859–60; Doe 2011, p. 62). Moreover, proportionality stricto sensu *"leads to a weighing between competing values to assess which value should prevail"* (Pirker 2013, p. 31) and *"requires a balancing of the benefits gained by the public and the harm caused to the constitutional right through the use of the means selected by law to obtain the proper purpose"* (Barak 2012, p. 340). The preferential interpretive of the principle of "practical harmony" (praktische Konkordanz) can be found in this last phase (parametre c) dictating the composition or at least the coexistence and balancing of the opposing constitutional interests (Scaccia 2019, p. 7).

The implementation of these evaluation techniques is necessary in order *"to balance the freedom of religion against competing freedoms that are also constitutionally protected, or against objective goals that have constitutional status. Balancing is not calculable but controlled"* (Engel 2011). Therefore, what should be done in this particular case when two obligations of the State: on the one hand, the assurance of religious worship, and on the other hand, the protection of public health, should simultaneously be serviced despite fighting each other? Based on the constitutional law it is accepted that *"an abstract manner of hierarchy between constitutional rights does not exist and in case that one conflicts another, they must be weighted in the manner of the specific actual circumstances existing each time. This is the right choice. However, when not only the health but even the citizens' life is put in danger, it is obvious that the protection of human life has increased weight in the aforementioned procedure. Because, ultimately, the existence of life consists the prerequisite for the conduction of all human rights"* (Vlachopoulos 2020).

## 5. The Relevant Case-Law

At primary stages of their implementation, the individual regulations of the emergency legislation, especially regarding the restrictions imposed to the exercise of the right of religious worship, were put under judicial judgement whose duty was to examine whether these restrictions exceeded the terms of proportionality.

In that context, the annulment of the ministerial decision no. 2867 was put into discussion by the Administrative Court of First Instance of Athens under the justification that the constitutionally regulated principle of proportionality is primarily infringed *"based on the fact, that the complete suspension of the conduction of the Mass in all Holy Temples and Monasteries should not be the only way to effectively protect the public health"*. In the context of these objections, the Court issued the decree no. 342/2020, which determined that according to the written case-law, temporary precautionary measures were taken in order to protect the public health and were not imposed as individual sanctions but rather as restrictions to the collective conduction of an individual right. Thus, the protection of the outweighing public health imposed the temporary restriction of conducting religious activities in accordance with the principle of proportionality.

The decision of the Commission of the Council of State, which was called to resolve the ongoing dispute, was towards the same direction. Similarly, in accordance with decision no. 49/2020, it was decided that the temporary nature of the measures and their reasonable duration combined with the lack of other measures that could immediately be taken for the effective protection of public health, due to the existence of overriding reasons of public interest, deemed the suspension of their application prohibitive.

The same rationale is also adopted by the decision no. 60/2020 of the aforementioned Council (Nomokanonika 2020, vol. 2, pp. 108–12, see similarly Council of State decision no. 99/2020), where it is supplemented that the documentation of the necessity of the measures taken to address the pandemic consists of a heavy responsibility of the Administration in accordance with the principle of proportionality, in view of their particularly restrictive nature regarding the exercise of fundamental rights, such as religious freedom and especially the freedom of worship. In fact, the fulfillment of this responsibility is subject to the corresponding judicial review that will necessarily take into account the time passed since the first need of adoption of the emergency measures for the prevention of the COVID-19 spread. Therefore, it is clear that the decision of the Council of State regarding the compatibility of the measures taken according to the principle of proportionality will not necessarily remain unchanged in the future as, amongst others, their short duration constitutes the main objective.

The same view is also expressed by the relevant decree no. 1 BvQ 28/20 on the 10th of April 2020 of the Federal Constitutional Court of Germany[7] (Bundesverfassungsgericht). In particular, with this decree and after having evaluated the request to provide temporary judicial protection against the prohibition of conducting holy activities during Easter, the Court decided that the prohibition of religious gatherings during the pandemic is legal as the protection of life is considered the ultimate goal, despite the fact that the prohibition constitutes an exceptionally severe interference in the religious freedom, especially during Easter, when the religious life of Christians is at its peak.

Nonetheless, the final paragraph of the Court (no. 14) according to which "*every extension of this temporary measure must be imposed to rigorous assessment regarding its proportionality while also taking into consideration the present conditions*" appears to be of paramount importance. In other words, the principle of proportionality should be strictly imposed due to the intensive interference in the public's religious worship. In practice, this means that the prohibition of religious gatherings must constantly be reevaluated based on the latest data about the progress of the spread of the virus as well as the durability of the health-care system, in order to determine whether the prohibition of religious gatherings can be replaced with less excessively restrictive measures (Vlachopoulos 2020, pp. 54–55).

In the same context, a recent decision of the Supreme Court of the United States must be mentioned (*Roman Catholic Diocese of Brooklyn, New York v. Andrew M. Cuomo, Governor of New York*, 592 U. S. ___ [2020]), which ruled that:

"Members of this Court are not public health experts, and we should respect the judgment of those with special expertise and responsibility in this area. But even in a pandemic, the Constitution cannot be put away and forgotten. The restrictions at issue here, by effectively barring many from attending religious services, strike at the very heart of the First Amendment's guarantee of religious liberty. Before allowing this to occur, we have a duty to conduct a serious examination of the need for such a drastic measure".

Similarly, on 5 February 2021, the US Supreme Court issued a pair of orders that overruled as unconstitutional California state restrictions that completely barred churches from meeting indoors but left in place percentage capacity limitations and bans on singing

---

7　See the decision available online: https://www.bundesverfassungsgericht.de/SharedDocs/Entscheidungen/DE/2020/04/qk20200410_1bvq002820.html.

or chanting (*South Bay United Pentecostal Church, et al., v. Gavin Newsom, Governor of California, et al.*, 592 U. S. ____ [2021])[8]. As Justice Gorsuch observed[9]:

"Since the arrival of COVID–19, California has openly imposed more stringent regulations on religious institutions than on many businesses [ . . . ] Apparently, California is the only State in the country that has gone so far as to ban *all* indoor religious services [ . . . ] Regulations like these violate the First Amendment unless the State can show they are the least restrictive means of achieving a compelling government interest [ . . . ] Of course we are not scientists, but neither may we abandon the field when government officials with experts in tow seek to infringe a constitutionally protected liberty. The whole point of strict scrutiny is to test the government's assertions, and our precedents make plain that it has always been a demanding and rarely satisfied standard. Even in times of crisis—perhaps *especially* in times of crisis—we have a duty to hold governments to the Constitution".

Additionally, at this point, it has to be indicated that the Administration, in accordance with its regulatory competence following a reasoned opinion from a specialized scientific committee, has the discretionary power to choose the appropriate, necessary, and as mild as possible measures, where relevant, for the protection of public health and definitely not the obligation to impose the prohibition of specific activities. Thus, it was determined by the Council of State, via its decree no. 161/2020, that by not prohibiting specific religious activities, such as the Sacrament, the competent minister does not omit a required legal action, as the evaluation of the purpose of proceeding or not to such kind of action is at the discretion of the substantive judgement of the Administration which is not under the control of the Court.

In the same context, the Administrative Court of First Instance in Larisa decided on objections raised in favor of the annulment of the ministerial order no. 50451 which had totally prohibited the litanies[10] outside the places of worship. The applicants supported that the general prohibition of the litanies during August constituted an interference in the internal affairs of the Church and opposed the principle of proportionality, as it nullified the minimum protection of the right of religious freedom, considering the fact that they were deprived of the conduction of the processions as prescribed by the worshipping practices of the Church, without an existing imperative reason of public interest, as the country was not subject to general curfew.

Additionally, via decree no. 17/2020 (see also decree no 1083/2020 of Administrative Court of First Instance, Athens) and by essentially repeating the reasoning and the legal basis of the decree no. 342/2020 of the Administrative Court of First Instance of Athens which proceeded it, the Court accepted that in view of the temporary nature of the measures there is no question of the State interfering in the internal affairs of the Church (interna corporis). This is so as the processions were not abolished by the contested ministerial decision perpetually but instead their conduction was suspended for a particularly short-term period of time (August of 2020) as determined by the equivalent magnitude of restriction of the freedom of worship in order to ensure the superior value public health of those living in the Greek territory.

Following its no. 49, 60 and 99/2020 decrees[11] which were issued in the context of temporary judicial protection, the Council of State was called at the end of May 2020 to reach a decision regarding the requests of annulment of those ministerial decisions, which temporarily abolished liturgies and holy activities in all places of religious worship without exemption. However, because the validity of the contested ministerial decisions had already expired as the conversations regarding the matter were taking place, the Court

---

8　For an overall approach of US courts' caselaw regarding the restrictions on religious liberty due to coronavirus pandemic, see Brian J. Buchanan, *Covid-19 and the First Amendment: A running report* (February 9), available online: https://www.mtsu.edu/first-amendment/post/613/covid-19 -and-the-first-amendment-a-running-report-dec-18 (accessed on 10 February 2021).

9　Joined by Justices Thomas and Alito.

10　A litany is defined as the religious ceremony with vows and hymns and with the participation of the clergy and the public in a procession as invocation to God in cases of an act of God or to honor a Saint, etc.

11　See also Council of State decision no 2/2021.

decided (see decree no. 1294-1296/2020) with a marginal majority and possibly via a formalistic approach in favor of no need to adjudicate.

More specifically, the Council of State determined that the applicants had no particular legal interest in the continuation of the proceedings that consisted of the moral damages caused by the contested ministerial decisions affecting the core of their right of religious freedom in the form of religious worship. The decision was based on the rationale that the recovery from the psychological and emotional ordeal, possibly caused by their separation from their religious community, could have been achieved by executing relevant action, without being necessary to issue relevant judgment annulling. Thus, the chance for the pleas to be objectively investigated by the Court was unfortunately lost.

## 6. Conclusions

There is no doubt that the measures undertaken have inevitably encroached on rights and freedoms which are an integral and necessary part of a democratic society governed by the rule of law. However, it is also accepted by the case-law, that *"Greece [ . . . ] is now a European paradigm of effectively dealing with the novel pandemic without deviations from constitutional order and protection of fundamental rights"* (Doudonis 2020). This statement is of great importance and despite the fact that frequently the *"Emergencies can be used as cover in for profoundly damaging changes to the constitutional limits and restrictions on the exercise of governmental power"* (Cormacain 2020). In that context, it should be accepted that the State utilized the opportunities provided by the Constitution and the European Convention of Human Rights regarding the management of a "health related state of emergency" to their full extent, by "playing" with their limits without surpassing them in general (Sotirelis 2020).

However, as it has already been highlighted, *"the saga of the (Greek) coronavirus crisis-law is, like everywhere, utterly reduced to the proportionality of the exceptional measures of the (Greek) State, but its moral and political implications seem far broader and ambiguous"* (Karavokyris 2020). Thus, we should not forget that *"Crises require decisive government action, but governments often use times of crises to encroach on individual freedoms or target minority groups long after the crisis has passed"* (Manchin and Carr 2020). Thus, the greatest danger is not the implementation of the measures taken—this is something temporary—but the potential to become used to them; this could be fatal. This is because *"the constitutional rights are really valuable. If at this moment we accept the temporary restriction of them in order to continue their exercise, this does not mean that it is allowed to get used to their loss"* (Vlachopoulos 2020, p. 23). As a conclusion, it is crucial to put a specific emphasis on the Council of Europe's declaration: *"The virus is destroying many lives and much else of what is very dear to us. We should not let it destroy our core values and free societies"*[12].

The emergency legislation was inevitably caused by the sudden appearance of the pandemic. In fact, the necessity of protecting the public health, which was under immediate danger, imposed measures to be taken by the government restricting other constitutionally established, important and legal values such as, in this case, religious freedom, particularly in the context of freedom of worship. These restrictive regulations were decided by the government in accordance with the results and guidance of expert scientists whose decisions cannot be assessed via lessons learned from common experience, and were hence obliged to validate them. It is true that, particularly for the Orthodox Church, the measures taken had a great impact mainly on Christians' collective religious freedom because they did not have, due to the restrictions imposed, the chance to take the Sacrament or to participate in several ceremonies as the churches were closed most of the time.

It is accepted that the regulation of the freedom of religious worship from the legal order also includes the establishment of specific restrictions. In this case, the refusal of the faithful crowd to conform to the laws cannot be justified by their religious beliefs. In fact,

---

12 Council of Europe, Information Documents SG/Inf(2020)11 (7 April 2020), *Respecting democracy, rule of law and human rights in the framework of the COVID-19 sanitary crisis. A toolkit for member states.* Available online: https://rm.coe.int/sg-inf-2020-11-respecting-democracy-rule-of-law-and-human-rights-in-th/16809e1f40 (accessed on 10 February 2021).

these laws must be of general application and ensure the protection of the important legal value which interests the general public and is protected by the Constitution (article 13 par. 4). The public health whose protection is imposed by the Constitution via the adoption of precautionary measures (article 21 par. 3) undoubtedly belongs amongst the last.

Of course, the permissible limits regarding the exercise of such a fundamental right are determined by the adequacy and necessity of the taken measure as well as by its proportionality to the desired result. In fact, the interference of the State in the religious freedom, in this particular case, must correspond to a "pressing social need", in accordance with the case-law of the European Court of Human Rights (*Svyato-Mykhaylivska Parafiya vs Ukraine*, 14.6.2007, recital 116) and therefore the term "necessary" does not have the same flexibility as the phrases "useful" or "desired".

According to the case law of the Greek courts, the principle of proportionality was not infringed by the restrictive measures regarding the freedom of worship, mostly because of the duration of the measures and their temporary nature in particular. Thus, it becomes obvious that as far as their content is concerned the prohibitive or restrictive measures related to the freedom of worship cannot remain unchanged, but they must be redesigned "*from time to time*" based on the renewed data regarding the pandemic, in order for the mildest and most adequate of them to be chosen. Although judicial review is limited to the temporary and revisable nature of the measures taken, it must be underlined that, at least indirectly, it takes into consideration that the burden caused by the restriction of freedom of religion does not exceed the positive outcome for the public interest (public health protection) caused by the above mentioned restriction (proportionality stricto sensu).

In the same context, it is difficult to support that the courts provided a blank cheque to the political authorities for the management of the COVID-19 pandemic. On the contrary, although they rejected the relevant requests of judicial protection, they kept a neutral attitude because they dictated the terms about the constitutional rescue and sustainability of the established measures.

Collaboration between the state and religious organizations (desirable in ordinary circumstances) becomes essential during a health emergency (Hill 2020). It is worth noting that the Orthodox Church and State relation in Greece was, in the main, a good paradigm.

**Funding:** This research received no external funding.

**Institutional Review Board Statement:** Not applicable.

**Informed Consent Statement:** Not applicable.

**Data Availability Statement:** Not applicable.

**Conflicts of Interest:** The author declares no conflict of interest.

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
