# Peer review of "The Right of Religious Freedom in Light of the Coronavirus Pandemic: The Greek Case"

_laws, 2021_

Round 1

Reviewer 1 Report

    This is an important, timely, and succinct paper that will be valued by professionals in the field. It merits, in my judgment, publication secondary to a few revisions. Below are changes I would suggest. 

    The paper uses italics in a manner unclear to me. Italics are used for some but not all quotations passim, and at other points also. Be sure to review this to ensure they are necessary. If so, an explanatory footnote would be helpful. 

     The paper's mention of the moral and political, and not merely legal, aspects of the issue is very helpful (57 and 15-19 in the abstract). However the paper does not fully deliver on what it promises in the abstract in terms of exploring the possibilities of governmental abuse and the potential for the public to grow used to limitations on important rights (15-17). Brief reference to these important points is made at the end, but they deserve either a fuller elaboration or removal of the topics altogether from the abstract and from line 57.  

     Supplying for those unfamiliar with Orthodoxy a bit more detail on the meaning and scope of what are labelled as Orthodox "religious duties" would be helpful to ensure an understanding of the full context, which is essential to the governing proportionality test.  (For example, can a priest perform per Orthodox "canon" law a eucharistic service only by himself? What is a verger?)

    In 105-116 assertion is made that the government has made the right decisions. These normative assessments, coming at this early stage of the paper, appear question-begging. It strikes me that evaluative claims should come only in the introduction (to telegraph to the reader the claims to come later) and in the last part, since evaluative claims must be assessed only after the full context is specified. So I would hold off on these evaluative claims in lines 105-116.

  Also, to ensure a fuller understanding of the totality of the circumstances in Greece, I would add a footnote explaining the "reprehensible dissonances" that the Church did engage in (line 114). 

   Lines 207-215 helpfully include a comparative perspective with US Supreme Court caselaw. Might that section be expanded into a fuller subsection drawing in a bit more comparative material from the United States?   

     Sections 4 and 5 are very helpful. 

In all, a fascinating and important paper.   

Reviewer 2 Report

The article would gain in doing a more detailed analysis of the principle of proportionality and its specificities in times of "emergency" (mostly in what regards the third sub-principle: proportionality stricto sensu), going, from a dogmatic point of view, a bit further than invoking the temporary and revisable nature of the measures taken. 

Also it would have been interesting to analyze in a more structured way the differential impact of the measures in the right(s) to individual and collective religious freedom and their several public manifestations. 
